# VAERS Vasculitis Adverse Events Retrospective Study: Etiology Model of Immune Complexes Activating Fc Receptors in Kawasaki Disease and Multisystem Inflammatory Syndromes

**DOI:** 10.3390/life14030353

**Published:** 2024-03-07

**Authors:** Darrell O. Ricke, Nora Smith

**Affiliations:** Lincoln Laboratory, Massachusetts Institute of Technology, Lexington, MA 02421, USA; nora.smith@ll.mit.edu

**Keywords:** Kawasaki disease, Multisystem Inflammatory Syndrome, Henoch–Schönlein purpura, MIS-C, MIS-A, MIS-N, MIS-V, Kawasaki disease shock syndrome, vaccines

## Abstract

Background: Vasculitis diseases include Kawasaki disease (KD), Kawasaki disease shock syndrome (KDSS), Multisystem Inflammatory Syndrome (MIS), Henoch–Schönlein purpura (HS), or IgA vasculitis, and additional vasculitis diseases. These diseases are often preceded by infections or immunizations. Disease incidence rates are higher in children than in adults. These diseases have been extensively studied, but understanding of the disease etiology remains to be established. Objective: Many studies have failed to demonstrate an association between vasculitis diseases and vaccination; this study examines possible associations. Methods: Herein, the Vaccine Adverse Event Reporting System (VAERS) database is retrospectively examined for associations between vasculitis diseases and immunizations. Results: For some vaccines, the number of rare cases of KD, MIS, and HS are higher than the background rates. These rare cases are predicted to occur in individuals with (1) genetic risk factors with (2) antibody titer levels above the primary immune response level. Herein, the model of humoral immune response antibodies bound to antigens (pathogen or vaccine) creating immune complexes is proposed. These immune complexes are proposed to bind Fc receptors on immune cells and platelets, resulting in cell activation and the release of inflammatory molecules including histamine and serotonin. Immune complexes and inflammatory molecules including serotonin and histamine likely trigger vasculitis. Elevated serotonin and possibly histamine drive initial vasoconstrictions, disrupting blood flow. Increased blood flow pressure from cardiac capillary vasoconstrictions is predicted to trigger coronary artery aneurysms (CAA) or lesions (CAL) in some patients. For KDSS and MIS patients, these cardiac capillary vasoconstrictions are predicted to result in ischemia followed by ventricular dysfunction. Ongoing ischemia can result in long-term cardiac damage. Cases associated with pathogens are likely to have persistent infections triggering disease onset. Conclusion: The proposed model of immune complexes driving disease initial disease etiology by Fc receptor activation of immune cells and platelets, resulting in elevated histamine and serotonin levels, is testable and is consistent with disease symptoms and current treatments.

## 1. Introduction

Kawasaki disease (KD), or mucocutaneous lymph node syndrome, and Multisystem Inflammatory Syndromes (MIS) are associated with the inflammation of medium-sized blood vessels (vasculitis). Henoch–Schönlein purpura (HS), or IgA vasculitis, is an IgA immune complex systemic vasculitis (inflammation of blood vessels) disease with palpable purpura (small, raised areas of bleeding underneath the skin). Vasculitis diseases have been linked to antibodies, circulating immune complexes of antibodies bound to antigens, infections, some vaccines, and genetic variants.

KD is a rare vasculitis disease of unknown etiology. KD results in a fever that typically lasts for more than five days and is unresponsive to paracetamol (acetaminophen) or ibuprofen. Additional common patient symptoms include extreme irritability, large lymph nodes in the neck, red eyes, very red lips, characteristic “strawberry tongue”, coronary artery aneurysms (~25%) or lesions, myocarditis, swollen lips with vertical cracking and bleeding, painful joints, and rashes in the genital area, lips, palms, and soles of the feet. The skin from the hands and feet may peel after recovery. KD frequently affects younger children (KD-C), aged 0 to 5, but can also appear in adults (KD-A). Fever associated with KD does not respond to normal fever treatments but does respond to high-dose aspirin. KDSS occurs with sustained systolic hypotension (decrease in blood pressure). Current treatments include high-dose aspirin, intravenous immunoglobulin, and sometimes the addition of a corticosteroid. Note that aspirin is normally counter-indicated for children due to the possible risk of Reye’s syndrome [1]. An increased proportion of KD patients with coronary artery aneurysms also had plasma fibrinogen (FG) alpha genotype Thr312Ala [2]. Genetic studies of KD susceptibility variants support the role of alterations in immune cell activation and the involvement of immune complex activation of immune cells and platelets [3,4]. The risk level for developing KD is likely higher for individuals with these genetic risk variants. KD is not thought to be contagious, though it may be caused by an infectious agent.

Multiple clinical studies have attempted to identify a causative infectious agent for KD. Clusters of KD cases frequently occur with a 4-to-6-week onset delay following pathogen outbreaks. Associations between KD and multiple pathogens have been reported as follows:Adenovirus [5,6];Human bocavirus [7];Coronavirus [6];Human coronavirus 229E [8];Human coronavirus (HCoV-NH) NL63 [9];Cytomegalovirus [10];Dengue [11,12];Enterovirus [6,13];Epstein–Barr virus [14];Human herpesvirus 6 [15];Human lymphotropic virus [16];Human rhinovirus [6];Influenza [17];Measles [18];Parvovirus B19 [19,20];Parainfluenza virus type 2 [21];Respiratory syncytial virus (RSV) [22];Rotavirus [23];Varicella zoster (chicken pox) [24,25];Torque teno virus [26];Staphylococcus aureus [27], and;Streptococcus [14,28].

KD has also been reported as a rare adverse event associated with vaccinations and vaccine combinations as follows:Bacillus Calmette-Gue’rin (BCG) vaccination [29].COVID-19 vaccine Vaxzevria (nonreplicating viral vector) [30].Diphtheria, tetanus, and acellular pertussis (DTaP or DTAP) [31].Hepatitis B [32].Influenza [31,33,34,35].Lanzhou lamb rotavirus vaccine (LLR) and freeze-dried live attenuated hepatitis A vaccine (HAV or HEPA) [36].Measles, mumps, and rubella (MMR) [31].Polysaccharide pneumococcal vaccine (Pneumo 23) [34].Pneumococcal conjugate vaccine (PCV or PNC) [31].Rotavirus [31,37].SARS-CoV-2 [38].Yellow fever vaccine [39].DTaP/Poliovirus vaccine inactivated (IPV)/Hepatitis B virus vaccine (HepB or HEP), [40].Prevnar 13 (PNC13 or PCV13).Rotarix [40].DTaP/IPV/Haemophilus B conjugate vaccine (Hib or HIBV)/PCV [41].DTaP/IPV/Hib; meningitis C; PCV [41].DTaP/IPV; MMR [41].DTaP-IPV/Hib and PCV13 [42].Hib; meningitis C; PCV; MMR [41].Measles/rubella (MR), varicella, pneumococcal [43].One report of an adult with both KD and (MIS-A) following second dose of the Pfizer SARS-CoV-2 vaccine [38].

Associations have also been made between KD and air pollutants such as carbon monoxide (CO), nitric oxide (NO), nitric dioxide (NO_2_), and nitrogen oxide (NOx) during pregnancy and childhood exposure [44]. In healthy blood vessels, the endothelium constitutively expresses nitric oxide synthase (NOSIII), which produce the vasoactive hormone nitric oxide (NO); in diseased blood vessels, vascular smooth muscle cells express inducible NOSII, resulting in the release of large amounts of NO [45]. For acute phase KD patients, neutrophils were identified as the major source of NO which decreases after intravenous immunoglobulin (IVIG) treatment [46]. NO relaxes blood vessels and inhibits platelet activation [47]. During the acute phase of KD, plasma hydrogen sulfide (H_2_S) levels significantly decreased and NO levels significantly increased (*p* < 0.01) [48]. Elevated levels of NO are associated with the development of coronary artery abnormalities in KD [49]. Elevated inducible NO and decreased H_2_S levels can predict the risk of coronary artery ectasia in KD patients [50]. H_2_S is involved in the regulation of vascular tone, blood pressure, and protection of the myocardium from ischemia-reperfusion injury (review [51]). A study of endothelial cells (ECs) found normal microvascular function in controls and after acute KD with EC injury confined to the endothelium of medium-sized arteries [52]. In contrast, plasma nitrate levels may not be associated with a higher risk of coronary artery lesions (CAL) [53]. A study of genetic polymorphisms of endothelial constitutive NOS (ecNOS) and inducible NOS (iNOS) genes did not find significant associations between CAL and KD [54]. The role of NO in KD pathogenesis was reviewed by Tsuge et al. [55]. Increased platelet activation markers and decreased levels of asymmetric dimethylarginine (ADMA) were found in KD patients compared to normal controls [56]. It is possible that elevated NO levels could be a marker for CAL. The etiology for KD remains unknown. While possible associations with multiple pathogens and vaccines have been reported in the literature (summarized above), no associations have been established and accepted by the medical community, which continues in its search to isolate the causative agent.

With overlapping symptoms with KD, Multisystem Inflammatory Syndrome (MIS) can affect different populations, including children (MIS-C), adults (MIS-A), neonates (MIS-N), and vaccine recipients (MIS-V). KD and KDSS, have many similar symptoms, such as inflammation of the heart, lungs, kidneys, brain, skin, eyes, and/or gastrointestinal tract. Like KD, MIS-C, MIS-A, and MIS-N are rare diseases. MIS-C, -A, and -N are associated with or follow SARS-CoV-2 infections. MIS-V is a rare potential occurrence after SARS-CoV-2 immunization in some vaccinees. With significant overlap with KD and KDSS symptoms, MIS-X (-C, -A, or -V) are characterized by persistent fever and possible acute abdominal pain with diarrhea or vomiting, muscle pain and general tiredness, inflamed blood vessels, low blood pressure, red eyes, rashes, enlarged lymph nodes, swollen hands and feet, “strawberry tongue”, coronary artery dilation to aneurysm, and various mental disturbances are possible. MIS-N patients often present with respiratory and cardiac symptoms, with only 18–20% presenting with fever [43,44]. Like KD, clusters of new cases can appear two to six weeks after local surges in SARS-CoV-2 infections.

Analogous to KD and MIS, Henoch–Schönlein purpura (HS), or IgA vasculitis, are associated with the formation of large circulating immune complexes with deposition in small blood vessels. Additional HS symptoms can include joint pain, abdominal pain, hematuria (blood in urine), and proteinuria (protein in urine); rare kidney involvement can proceed to chronic kidney disease. HS primarily occurs in children, with a higher frequency in males. Cases of HS have been reported after Epstein–Barr virus [57,58,59], Helicobacter pylori [60], parainfluenza [61,62], parvovirus B19 [63,64,65], Staphylococcus [62], Streptococcus [62], and Varicella zoster [66] infections, cytomegalovirus reactivation [67], and BCG therapy [68]. Case reports of HS have been reported following hepatitis A [69], influenza [70,71,72,73], MMR [74], meningitis C [75], rabies [76], and SARS-CoV-2 mRNA vaccination [77,78,79,80,81,82,83]. In contrast to these case reports, a systematic literature review reported no causal association between vaccination and KD or HS [84].

During both infection and vaccination, foreign proteins are introduced into the body. Antibodies bind these antigens, forming immune complexes. Immune complexes bind and activate immune cells and platelets with Fc receptors. Activated platelets release serotonin, histamine, and additional inflammatory molecules. Activated mast cells and other granulocytes also release histamine and other inflammatory molecules. Elevated serotonin (5-hydroxytryptamine) is associated with coronary artery disease and some cardiac diseases [85]; serotonin has several effects on the vascular wall, proliferation of smooth muscle cells, promotes thrombogenesis and mitogenesis [85]. Serotonin released from activated platelets is associated with vasoconstriction [85,86] and also vasodilation in the absence of endothelium damage [86]. Serotonin-specific effects can be blocked by the serotonin receptor antagonist ketanserin [85]. Fluoxetine, a serotonin reuptake inhibitor (SSRI), is associated with urticaria and angioedema in one case report [87]. Cardiac adverse events of the β-imanazolylethylamine derivative of histamine include altered blood-pressure, constriction of coronary arterioles, constriction of pulmonary arterioles, altered heart rate, and heart failure varying according to dose and animal species [88]. Histamine is also involved in cardiac arrhythmias [89] and cardiac adverse events associated with histamine intolerance (HIT) [90]. Elevated histamine and/or serotonin are likely associated with cardiac vasoconstrictions associated with KD, KDSS, and MIS.

This study retrospectively analyzes reports of adverse events for KD, MIS, HS, and vasculitis in the Vaccine Adverse Event Reporting System (VAERS). The etiology of these diseases is considered in the context of associations with multiple infectious pathogens and also multiple possible vaccine associations. The hypothesis that KD, KDSS, MIS-C, MIS-A, MIS-N, and MIS-V are associated with adverse reactions to immune complexes (antibodies bound to pathogen or vaccine proteins) with associated Fc receptor activation of immune cells and platelets, releasing serotonin, histamine, and other inflammatory molecules, is advanced. Likewise, immune complexes likely drive the other vasculitis diseases.

## 2. Materials and Methods

This is a retrospective analysis of the VAERS database from 1 January 1990 to 27 October 2023. The names of VAERS adverse events searched for were Henoch-Schonlein purpura, Kawasaki’s disease, Multisystem Inflammatory Syndrome, Multisystem Inflammatory Syndrome in adults, Multisystem Inflammatory Syndrome in children, and vasculitis. No patients with these adverse events were excluded. The Python program vaers_reports.py was developed for retrospective analysis of the VAERS data files VAERSDATA, VAERSSYMPTOMS, and VAERSVAX for the years 1990 to 2023 and NonDomestic.

## 3. Results

Adverse events occur at background population frequencies with the addition of any vaccine-associated adverse events. For adverse event, *X*, the reported numbers are diminished by reporting bias, $r_{day}^{X}$, as time (day) increases. Background population events (*b*) for individuals of the same age and a specific adverse event (*X*) are modeled as a constant, noted as $b_{age}^{X}$. The reporting bias (*r*) for a specific adverse event (*X*) is modeled as a different constant dependent upon the number of days since immunization, noted as $r_{day}^{X}$. For a population, *P*, and *n* days of data collection, the expected number of background adverse events can be modeled by the following Equation (1), where $r_{day}^{X}$ and $b_{age}^{X}$ are a series of constants for days and age, respectively:(1)$$Adverse\;Events\left(X\right)=\sum_{age=0}^{100}P_{age}\;\times \sum_{day=0}^{n}r_{day}^{X}\;\times \;b_{age}^{X}\;\;\;$$

The addition of any vaccine-specific, *V*, associated adverse events can be modeled with an additional age series of constants, $v_{age}^{X}$, in Equation (2):(2)$$Adverse\;Events\left(X,\;V\right)=\sum_{age=0}^{100}P_{age}\;\times \sum_{day=0}^{n}r_{day}^{X}\;\times \;{(v_{age}^{X}+b}_{age}^{X})$$

If the association of KD with vaccines only reflects the background population frequency rate, then the normalized frequencies for different vaccines should be similar with expected random variations. The normalized frequency of KD cases per 100,000 VAERS reports with symptoms is summarized in Table 1 for vaccines with five or more KD adverse event reports. The normalized frequencies per 100,000 VAERS reports varied widely from lower frequencies associated with the vaccines against Diphtheria and tetanus toxoids and the pertussis vaccine (DTP) (51 per 100,000) to the Meningococcal group b vaccine (MENB) (3262 per 100,000), with only seven VAERS vaccine codes having frequencies greater than 1000 per 100,000 (Table 1). The large differences in observed normalized frequencies in Table 1 are inconsistent with expected background variations, as modeled by Equation (1). Normalized frequencies of 93 and above were evaluated with a chi-squared test (*p* < 0.00001) compared to background KD frequency of (9 to) 20 per 100,000 per year [91]; the results are still significant after Bonferroni multiple testing correction (*p* < 0.00167) for 29 of 30 statistical tests (normalized frequencies of 72 and above). Note that the normalized frequencies in Table 1 represent underestimates after considerations for partial year for VAERS reports (120 days), reporting bias, etc.

The reported days of onset for KD cases post vaccination are summarized in Table 2 for nine vaccines. Day 0 and 1 had the highest numbers of KD reports (Table 2). The data in Table 2 likely reflect underreporting due to reporting bias as modeled in Equation (2).

The clinical symptoms for multiple diseases, including KD, KDSS, MIS, mast cell activation syndrome (MCAS), and type III hypersensitivity share overlaps as summarized in Table 3. Many of the symptoms could be possibly associated with elevated histamine levels and also serotonin; activated platelets are likely source for both, and granulocytes including mast cells are a possible additional source of histamine.

The age of onset for KD, MIS-C, and MIS are summarized in Table 4. The normalized frequencies of KD, MIS-C, MIS, HS, and vasculitis are summarized by age when receiving COVID-19 vaccines in Table 4. For KD, infants less than one year of age had the highest number of reports (689 reports), followed by one-year-old infants (169 reports) (Table 4). Likewise, vasculitis had the highest number of reports for infants less than one year of age (169 reports) and one-year-old infants (116 reports) (Table 4). Vasculitis reports in infants are driven by the following vaccines: 6VAX-F, DTAP, HEP, HEPA, HIBV, IPV, MMR, PNC, PNC13, RV1 (Rotavirus vaccine, live, oral), RV5 (Rotavirus vaccine, live, oral, pentavalent), and VARCEL. The Henoch–Schönlein purpura adverse events peak for infant vaccinations (104 and 80 for infants aged <1 and <2, respectively) and also preschool vaccinations at age 4 (81 reports) and 5 (96 reports); for HS, increases for children aged 4 and 5 years old are driven by MMR, DTAP, MMRV (measles, mumps, rubella and varicella vaccine live), DTAPIPV (Diphtheria and tetanus toxoids and acellular pertussis vaccine and inactivated poliovirus vaccine), and VARCEL reports in VAERS (Table 4).

The age of onset and normalized frequency per 100,000 vaccinations with symptoms associated with adverse events for COVID-19 vaccination, namely KD, MIS, MIS-A, and MIS-C, are summarized in Table 5; the frequencies from age 1 to 13 are higher (mean 391.5, SD 204.9) with observed lower frequencies for ages 14 to 17 (mean 142.5, SD 19.1) followed by much lower rates for adults aged 18 to 30 (mean 19.9, SD 15.1).

## 4. Discussion

Authors: multiple hypotheses can be proposed for adverse events for Kawasaki disease:

**Hypothesis** **1.**
*There is no association of Kawasaki disease with vaccination. This is the current general medical consensus.*


**Hypothesis** **2.**
*Kawasaki disease occurs in a subset of individuals with immune complexes from persistent infections or immunization activating immune cells and platelets with the risk level increased by relevant genetic variants. The number of adverse events observed for each vaccine may be correlated with vaccine reactogenicity level or with other vaccine-specific attributes including possible manufacturing contaminations.*


The large disparities between the normalized frequency of Kawasaki disease per 100,000 reports with any symptom across multiple vaccines in VAERS, summarized in Table 1, is inconsistent with these events representing only background adverse events, as modeled by Equation (1); hence, Hypothesis 1 is rejected. The data shown in Table 1 are consistent with modeling by Equation (2) and Hypothesis 2; other possible hypotheses may be consistent with the data observed in Table 1.

### 4.1. Kawasaki Disease and Multisystem Inflammatory Syndromes Etiology Model

It has been previously proposed that KD is associated with immune complexes [4,92]. KD has been characterized as a triphasic illness with the first phase (feverish phase) characterized by high fever, mucocutaneous manifestations, lymphadenopathy, and normal platelet count. The second phase (subacute phase) includes a dramatic rise in platelet count, may include immune complex vasculitis, and also involves desquamation of the hands and feet and sometimes coronary artery aneurysms (CAA). The third phase is the convalescent phase, where the platelet count decreases and immune complexes become undetectable [92]. Platelet counts are typically elevated by the second week of illness [93]. Immune-complex-induced platelet aggregation correlates with IgG and IgA titers [92]. In a murine KD model, platelets promoted vascular inflammation via the formation of monocyte–platelet aggregates (MPAs) and exacerbated the development of cardiovascular lesions [94]. In KD, platelets and activated monocytes can result in Kawasaki disease complicated with macrophage activation syndrome (KD-MAS) [95]. Immune complex activation of platelets is a key step in the proposed KD and MIS disease etiology.

The disease symptoms for KD and MIS combined with disease characteristics, associations with multiple pathogens, associations with multiple vaccines, and unusual treatments point towards a candidate etiology of immune complexes activating immune cells and platelets via Fc receptor binding; this results in the release of inflammatory molecules including histamine and serotonin (Figure 1). This model proposes that antibody levels that are higher than primary immune response levels binding to either infectious pathogen or vaccine proteins, creating immune complexes, are circulating and activating immune responses by binding to Fc receptors on immune cells and platelets, including granulocytes and mast cells. Vasculitis, rash, and fever are associated with Type III hypersensitivity-like responses to immune complexes. The activation of platelets and likely granulocytes, including mast cells, induces the release of high levels of histamine, serotonin, and other inflammatory molecules associated with mast cell activation syndrome (MCAS) symptoms, including fever, rash, diarrhea, nausea, vomiting, red eyes, headaches, migraines, palpitation, arrhythmia, etc. Elevated levels of serotonin are associated with vasoconstrictions [85]. Contractions of cardiac capillary pericyte cells in response to high levels of histamine have been previously proposed to cause (pressure-induced) coronary artery aneurysms, myocarditis, and pericarditis [96]. Pericytes, attached to the surface of capillaries, play an important role in local blood flow [97]. Pericytes are also involved in innate immune responses [98]. Pericytes also mediate coronary no-reflow after myocardial ischemia [99]. Pericyte loss has been correlated with microaneurysm size in diabetic retinopathy [100]. The SARS-CoV-2 Spike protein disrupts human cardiac pericytes function through CD147 receptor-mediated signaling [101] with possible relevance in MIS. Mast cells and eosinophils are known to create feedback loops. The eosinophil-to-lymphocyte ratio is a useful diagnostic for KD [102]. Immune responses to infectious pathogens are likely associated with overlapping symptoms.

Rare KD cases frequently appear roughly 4 to 6 weeks following pathogen outbreaks or SARS-CoV-2 infection for MIS-C, MIS-A, and MIS-N. This etiology model proposes that primary immune response antibody titer levels are insufficient to trigger KD and MIS due to low affinity for IgG1 antibodies by Fc receptors on immune cells and platelets. Ongoing infections will continue antibody responses above the primary immune response levels and may be a component in the delayed onset of KD and MIS following pathogen outbreaks. KD and MIS are not considered contagious. For these cases associated with pathogen outbreaks, this etiology model proposes that the patient has an ongoing pathogen infection, perhaps gastrointestinal (based upon MIS-C and MIS-A cases [103,104,105,106]). Prior exposure to the antibody antigen via previous infection or vaccination is an alternative to ongoing infection with secondary antibody response antibody titer levels that are significantly higher than primary antibody response levels. Rare KD and MIS cases appear in neonates (KD-N) [107,108,109] and MIS-N [110,111,112]. This etiology model proposes that these neonates likely have higher antibody titer levels due to a combination of neonate antibody responses combined with maternally acquired antibodies transferred during pregnancy [110].

### 4.2. KD and MIS Treatments and Etiology Model

The current KD and MIS treatments are high-dose aspirin, IVIG, and, sometimes, a corticosteroid. IVIG treatment within 7 days of illness onset lowers the risk of the patient developing coronary artery lesions and cardiac sequelae [113]. Treatment with aspirin also lowers the risk of developing coronary artery lesions [114]. KD patients presenting with fever did not respond to normal fever treatments. The proposed etiology model provides an explanation for mechanisms of actions for current treatments. The current treatments all target immune complex binding to antibody-heavy chains via Fc receptors (IVIG), mast cell stabilization (high-dose aspirin), or immune modulation (corticosteroid) (Figure 2). We propose that the efficacy of IVIG treatment is likely due to competitive binding to Fc receptors, reducing the binding of immune complexes to immune cells and platelets. Aspirin is also an inhibitor of the cyclooxygenase-1 (COX-1) and -2 (COX-2) pathways, with both pathways involved in inflammation immune responses. Plasma exchange, when used to treat KD patients [108], also reduces circulating immune complexes. Additional candidate treatments targeting mast cells, antibody binding to Fc receptors, and serotonin receptors may be worth evaluating in approved clinical studies.

### 4.3. Kawasaki Disease and Vaccinations (KD-V)

Calculating normalized frequencies of KD cases per 100,000 vaccinations with symptoms enables comparisons of frequencies across multiple vaccines. Some vaccines have a much higher normalized frequency (e.g., MENB [Meningococcal group b vaccine, rDNA absorbed], TYP [Typhoid vaccine], BCG, DTPIPV [Diphtheria and tetanus toxoids, pediatric and inactivated poliovirus vaccine], 6VAX-F [Diphtheria and tetanus toxoids and acellular pertussis adsorbed and inactivated poliovirus and hepatitis B and haemophilus B conjugate vaccine], and MEN [Meningococcal polysaccharide vaccine]) compared to vaccines with lower normalized frequencies (e.g., DTP, VARCEL [Varivax-varicella virus live], and FLU3 [Influenza virus vaccine, trivalent]), as shown in Table 1; these vaccines all include components from pathogenic bacteria. Prior exposure to pathogens, previous vaccine doses received, vaccine reactogenicity level, unknown characteristics of vaccines, excipients, or even possible contaminants might impact the observed differences. It is clear from Table 1, showing normalized frequencies, that higher normalized frequencies observed do not simply reflect background population occurrences. Likewise, genetic risk variants alone cannot account for the disparities observed between different vaccine normalized frequencies. The data reported to VAERS are negatively impacted by reporting bias with expected decreased probability of reporting as the number of days post immunization increases. Table 2 summarizes the day of KD onset post vaccination. Days 0 and 1 reflect the highest days for KD reports; this may simply reflect reporting bias. Alternatively, this may reflect activation of memory immune cells from previous antigen exposures.

### 4.4. Nitric Oxide (NO) and Coronary Artery Lesions (CAL)

This model predicts that coronary artery dilations and aneurysms are pressure-induced by cardiac capillary vasoconstrictions; hence, the risk level for CAL may be highest in KD-N [107,108,109] and MIS-N [111,112] patients. This simple model shows that artery resilience to increased pressure improves with age, with neonates having the lowest resilience to increased pressure levels. The risk level for CAL may decrease with increased age; ischemia from proposed cardiac capillary vasoconstrictions likely results in ventricular dysfunction observed in KDSS, MIS-C, and MIS-A patients. Cardiac capillary vasoconstrictions and resulting ischemia can also result in hypotension, as seen in KDSS patients. For patients developing CAL, this model predicts that the first step is pressure-induced dilation of the coronary artery, triggering cellular injury signaling that attracts the infiltration of immune response cells. At the sites of pressure-induced injuries, the induction of NO has been observed [46,49]. These same injury response signals were not observed in KD patient control internal mammary arteries [115]. Dilation and aneurysm injuries are specific to the coronary artery due to the closest associations to the predicted cardiac capillary vasoconstrictions elevating the risk of experiencing pressure-induced injuries [96]. Hence, elevated NO and, also, decreased H_2_S levels are likely indicators of CAL injuries.

### 4.5. Multisystem Inflammatory Syndromes (MIS-C, MIS-A, MIS-N, and MIS-V)

With parallels to KD and KDSS, Multisystem Inflammatory Syndrome appeared in children (MIS-C), adults (MIS-A), and neonates (MIS-N) associated with SARS-CoV-2 infections [105,106,111,112,116] and COVID-19 vaccines (MIS-V) [117,118]. The age of onset of MIS-C in children is older than for KD [104,105,106]. KD and MIS are currently considered to be distinct due to differences in patient age demographics and additional MIS-X symptoms (e.g., ventricle dysfunction, gastrointestinal, and neurological) [119]. Complement activation is seen in some MIS-C children with rapid improvement after IVIG treatment; however, this was not associated with detectable immune complexes [120]. Roughly half of MIS-C patients suddenly developed cardiogenic shock requiring intensive care unit (ICU) admission in 50% [116] to 80% [105,106] of patients. For MIS-C, sustained levels of inflammatory macrophage-activating, Fc receptor-binding antibodies were selectively maintained in severe disease [121]. In a review study, the majority of MIS-C patients are reported as SARS-CoV-2 IgG-positive [122]. In contrast, SARS-CoV-2-specific IgA antibody responses linked to neutrophil activation in severe MIS-A disease [121]. The age of KD, MIS-C, and combined ages are compared in Table 4. For KD-V and vasculitis, the majority of the cases are associated with infant vaccinations (Table 4). For vasculitis, reports associated with the vaccines COVID-19, HEP, HPV2, and HPV4 predominate for teenagers. In contrast, cases of MIS-V (recorded in VAERS as MIS-C) are primarily children aged 5 to 17 years, with fewer COVID-19 vaccinations administered to infants.

Here, we propose that KD, KDSS, and MIS-X all share the same etiology model driven by immune complexes activating immune cells and platelets. The main difference between MIS-C and KDSS is where SARS-CoV-2 was identified as the infectious agent [123]. KD is associated with CAL and long-term cardiovascular sequelae, while MIS-C presents as more intense Inflammatory Syndrome, myocardial dysfunction, and cardiogenic shock [124]. In a retrospective review of 395 MIS-C and 69 KD patients, MIS-C patients presented with gastrointestinal (80%), cardiovascular (74%), and respiratory (52%) symptoms, while KD patients presented with dermatological (99% vs. 68%) and mucosal changes (94% vs. 64%), and cervical lymph node swelling (51% vs. 34%) [125]. Additional MIS-X symptoms may derive from ongoing SARS-CoV-2 infections and differences in the age demographics for current cases. Deep immune profiling identified activated macrophages, neutrophils, B-plasmablasts, and CD8+ T cells in MIS-C patients with activation largely independent of anti-SARS-CoV-2 humoral immune response [120]; rapid improvement of MIS-C was driven by decreased activation of complement following IVIG treatment [120]. Table 5 suggests that the KD and MIS-X risks levels may start decreasing at age 14 with low risk levels for adults.

### 4.6. Henoch–Schönlein Purpura and Other Vasculitis Diseases

IgA immune complexes cause Henoch–Schönlein purpura, and immune complexes cause other vasculitis diseases. For diseases following infections, it is proposed that persistent infections provide antigens for the immune complexes. For immunization-associated onset of these vasculitis diseases, the vaccine protein provides the antigen. Many adverse events as a result of HS-C, KD, and vasculitis appear to be associated with childhood immunizations (Table 4).

### 4.7. Study Limitations

The VAERS database collects only a small subset of adverse events experienced by vaccinees. Any reporting biases or exclusion of adverse events would perturb the accuracy of VAERS representing the immunization population (note: reporting bias was anticipated and modeled in this study).

### 4.8. Study Recommendations

This study proposes that immune complexes binding to Fc receptors drive the etiology of both KD and MIS. Many of the disease symptoms are consistent with predicted elevated levels of histamine and/or serotonin. Evaluations of adjunctive treatments targeting elevated histamine or serotonin levels are candidates for evaluation in approved clinical studies. Early treatments may reduce the risk levels of CAL in KD patients and ventricular dysfunction in MIS patients. KD and MIS cases not associated with immunizations may have undiagnosed persistent infections for which appropriate treatments can be considered.

## 5. Conclusions

This retrospective study identified candidate associations between KD and 29 vaccines. This study, combined with other reported pathogen and vaccine associations, supports the hypothesis that KD is associated with multiple pathogens and vaccinations, with possibly implication of immune complexes. The etiology model of immune complexes binding Fc receptors and activating immune cells and platelets, driving both Kawasaki disease and Multisystem Inflammatory Syndrome, was advanced. Disease onset is conjectured to occur in the context of persistent infections or antibody titer levels higher than primary immune response levels. Histamine, serotonin, and inflammatory molecules are proposed to induce cardiac capillary vasoconstrictions resulting in pressure-induced coronary artery aneurysms, ventricular dysfunction, and other cardiac adverse events. Rare vaccine associations were observed for multiple vaccines with higher normalized frequencies for some bacterial vaccines; for Kawasaki disease, multiple of these associations are with live vaccines. Childhood immunizations are age-associated with Kawasaki disease onset. It can be inferred that the risk level of coronary artery dilations and aneurysms likely decreases with age from the model for pressure inducement from cardiac capillary vasoconstrictions. Persistent infections with higher antibody titer levels combined with genetic risk factors are likely causes of Kawasaki disease, Multisystem Inflammatory Syndromes, and Henoch–Schönlein purpura; vaccines are an alternate source of antigens or attenuated pathogens that can also drive the formation of immune complexes.

## Figures and Tables

**Figure 1 life-14-00353-f001:**
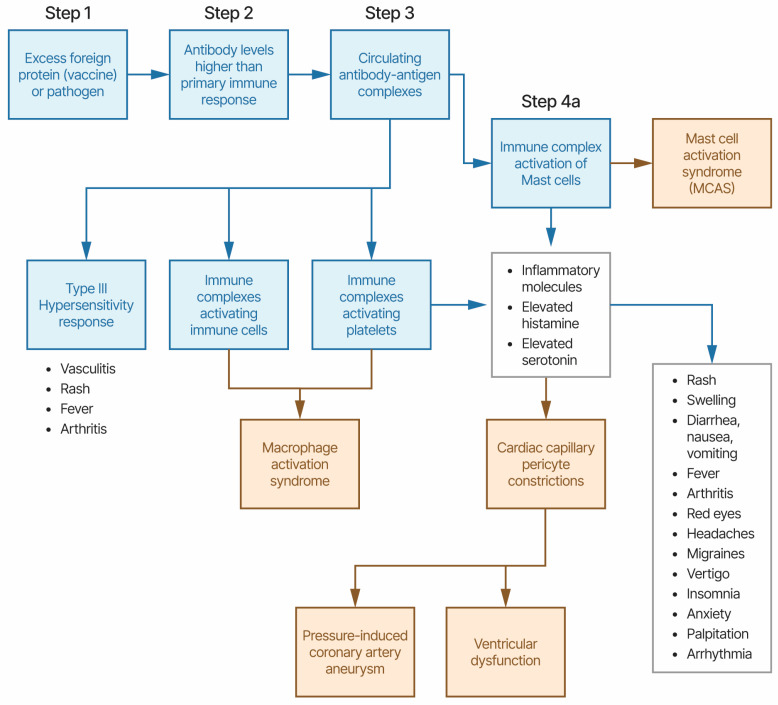
Etiology model of Kawasaki disease and MIS-X. The symptoms present in boxes shaded blue are likely in most patients; those present in boxes shaded yellow are likely in a subset of patients.

**Figure 2 life-14-00353-f002:**
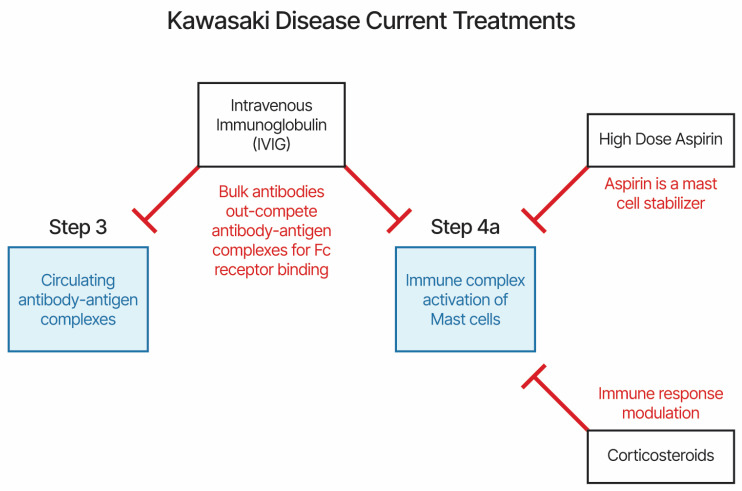
Current treatments for Kawasaki disease and MIS-X.

**Table 1 life-14-00353-t001:** Kawasaki disease adverse event reports with normalized frequency per 100,000 patients with reported symptoms in VAERS; data from 1990 to 27 October 2023.

Vaccine Code	Vaccine Name	Vaccinations Age 0–5	All	Frequency
6VAX-F	Diphtheria and tetanus toxoids and acellular pertussis adsorbed and inactivated poliovirus and the hepatitis B and haemophilus B conjugate vaccine	4092	68	1662
BCG	Bacillus Calmette–Guerin vaccine	241	5	2075
COVID-19	Coronavirus 2019 vaccine	7267	54	743
DTAP	Diphtheria and tetanus toxoids and acellular pertussis vaccine	55,155	57	103
DTAPHEPBIP	Diphtheria and tetanus toxoids and acellular pertussis vaccine and hepatitis B and inactivated poliovirus vaccine	11,606	41	353
DTAPIPV	Diphtheria and tetanus toxoids and acellular pertussis vaccine and inactivated poliovirus vaccine	10,939	15	137
DTAPIPVHIB	Diphtheria and tetanus toxoids and acellular pertussis vaccine and inactivated poliovirus vaccine and haemophilus B conjugate vaccine	10,861	55	506
DTP	Diphtheria and tetanus toxoids and pertussis vaccine	21,679	11	51
DTPIPV	Diphtheria and tetanus toxoids, pediatric and inactivated poliovirus vaccine	504	10	1984
FLU3	Influenza virus vaccine, trivalent	9145	7	77
FLU4	Influenza virus vaccine, quadrivalent	5270	14	266
FLUX	Influenza virus vaccine, unknown manufacturer	2181	18	825
HBHEPB	Haemophilus B conjugate vaccine and hepatitis B	5236	8	153
HEP	Hepatitis B virus vaccine	20,648	58	281
HEPA	Hepatitis A vaccine	18,061	30	166
HIBV	Haemophilus B conjugate vaccine	52,928	126	238
IPV	Poliovirus vaccine inactivated	36,648	44	120
MEN	Meningococcal polysaccharide vaccine	1239	16	1291
MENB	Meningococcal group b vaccine, rDNA absorbed	2790	91	3262
MMR	Measles, mumps and rubella virus vaccine, live	59,053	64	108
MMRV	Measles, mumps, rubella and varicella vaccine live	11,873	11	93
PNC	Pneumococcal 7-valent conjugate vaccine	26,198	107	408
PNC13	Pneumococcal 13-valent conjugate vaccine	23,510	160	681
PPV	Pneumococcal vaccine, polyvalent	4573	16	350
RV1	Rotavirus vaccine, live, oral	7858	79	1005
RV5	Rotavirus vaccine, live, oral, pentavalent	16,921	104	615
RVX	Rotavirus (no brand name)	1398	6	429
TYP	Typhoid vaccine	233	6	2575
UNK	Unknown vaccine type	2840	24	845
VARCEL	Varivax-varicella virus live	43,062	31	72

**Table 2 life-14-00353-t002:** Kawasaki disease onset post vaccination from VAERS; data from 1990 to 27 October 2023.

Onset Day	PNC13 *	HIBV	PNC	RV5	MENB	RV1	6VAX-F	MMR	HEP
Blank	41	31	20	16	39	23	15	24	9
0	23	17	19	11	13	4	15	3	5
1	23	28	10	9	4	10	6	4	11
2	9	6	6	5	3	2	5	6	
3	7	4	5	5	1	4	2	1	5
4	9	1	1	5	3	4	2	3	1
5	5	5	2	3	1	2	1	2	2
6		2	1		2		1	2	3
7		2	1			1		3	1
8	4	1	2	2	1		2	3	
9	4	3	1	3	2	2	1		3
10	1			1	1				2
11	1	1	1	2		1		2	
12	2		2	1		3	2	1	2
13	2		1		3	2	2		2
14	4	1	4	3		5	1		
15	2		1	2	1	2		2	
16	1	1	1	2	1	2			2
17	2	1	1	1		1	1		1
18	1			1		1	1		
19		3	4	1	1				
20		1		1			2	1	
21		2	2	1				1	1
22	1			1				1	1
23	1			1	1	1	1		1
24				1					
25		1			1				
26	4	1		1		2	1		
27					1				
28	1		1			1	1	2	
29					1				
30	1	1	3	3		2	1		1

* Vaccine code names: PNC13 (Pneumococcal 13-valent conjugate vaccine), HIBV (Haemophilus B conjugate vaccine), PNC (Pneumococcal 7-valent conjugate vaccine), RV5 (Rotavirus vaccine, live, oral, pentavalent), MENB (Meningococcal group b vaccine, rDNA absorbed), RV1 (Rotavirus vaccine, live, oral), 6VAX-F (Diphtheria and tetanus toxoids and acellular pertussis adsorbed and inactivated poliovirus and hepatitis B and haemophilus B conjugate vaccine), MMR (Measles, mumps and rubella virus vaccine, live), and HEP (Hepatitis B virus vaccine).

**Table 3 life-14-00353-t003:** Symptom overlaps between Kawasaki disease, KDSS, MIS, and mast cell activation syndrome (MCAS).

Organ System	Symptoms	Kawasaki Disease	Kawasaki Disease Shock Syndrome	Multisystem Inflammatory Syndrome	Mast Cells Activation Syndrome	Type III Hypersensitivity
Respiratory	Swelling of lips	Yes	Yes	Yes		
	Swelling of tongue, strawberry tongue	Yes	Yes	Yes		
	Swelling of throat, eustachian tube, glottis	Yes	Yes			
	Enlarged lymph nodes/lymphadenopathy	Yes	Yes	Yes		
Fever	Persistent fever	Yes	Yes	Yes		Yes
Gastrointestinal	Diarrhea		Yes		Yes	
	Abdominal pain	Yes	Yes	Yes	Yes	
	Nausea	Yes	Yes	Yes	Yes	
	Emesis/vomiting	Yes	Yes	Yes		
Neurological	Headache/migraine		Yes	Yes	Yes	
	Itchy, watery, red/conjunctivitis	Yes	Yes	Yes	Yes	
Circulatory/cardiovascular	Chest pain	Yes	Yes	Yes	Yes	
	Vasculitis	Yes	Yes	Yes		Yes
	Coronary artery lesions	Yes	Yes	Yes		
	Myocarditis	Yes	Yes	Yes		
Integumentary (skin)	Pruritus (itchy skin)	Yes	Yes	Yes	Yes	
	Flushing/redness/erythema	Yes	Yes	Yes	Yes	
	Urticaria/hives/rash	Yes	Yes	Yes	Yes	
	Swelling of hands and feet	Yes	Yes	Yes		

**Table 4 life-14-00353-t004:** Age of onset for Kawasaki disease, MIS-C, Henoch–Schönlein purpura, and vasculitis adverse events in VAERS; data from 1990 to 27 October 2023.

Age	Kawasaki	MIS-C	KD, MIS, MIS-A, or MIS-C	Henoch–Schönlein Purpura	Vasculitis
N.A.	420	71	578	233	667
0	689	0	689	104	169
1	169	6	174	80	116
2	31	0	31	31	29
3	19	2	21	33	13
4	14	1	14	81	27
5	8	11	18	96	22
6	8	5	13	35	8
7	4	17	21	26	13
8	1	10	11	28	6
9	4	13	17	18	6
10	1	18	18	21	0
11	2	10	13	26	6
12	5	20	24	32	25
13	3	15	17	25	20
14	1	9	10	26	8
15	1	11	12	21	14
16	2	12	14	16	16
17	2	15	15	15	18
18	2	1	4	6	22
19	0	1	3	4	11
20	0	2	4	10	16

**Table 5 life-14-00353-t005:** COVID-19-associated adverse events for KD, MIS, MIS-A, or MIS-C normalized frequency per 100,000 cases with symptoms by age in VAERS; data from 1990 to 27 October 2023.

Age	KD, MIS, MIS-A, or MIS-C	Vaccinations	Frequency/100K
1	3	851	352
2	2	826	242
3	2	865	231
4	1	1071	93
5	11	2751	399
6	6	1921	312
7	19	2117	897
8	10	2194	455
9	16	2417	661
10	13	2687	483
11	11	3950	278
12	22	6136	358
13	16	4862	329
14	7	5306	131
15	11	6410	171
16	11	8329	132
17	14	10,274	136
18	3	7237	41
19	3	7497	40
20	4	8321	48
21	2	8716	22
22	3	9445	31
23	1	10,291	9
24	2	10,985	18
25	1	11,912	8
26	0	12,456	<8
27	1	13,197	7
28	2	13,813	14
29	1	14,700	7
30	0	16,084	<6

## Data Availability

Data are contained within the Appendix A.

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
