# Peer review of "VAERS Vasculitis Adverse Events Retrospective Study: Etiology Model of Immune Complexes Activating Fc Receptors in Kawasaki Disease and Multisystem Inflammatory Syndromes"

_life, 2024, doi:10.3390/life14030353_

Round 1
Reviewer 1 Report
Comments and Suggestions for Authors
Dear colleagues,
In this article authors have tried to study the etiology of Kawasaki disease and presented useful literature data.
Following are some major comments:
- The title of the article should be corrected. Publication type must be specified (meta-analisis, review or systematic review
- The structure of the abstract should be corrected and should include: background; objective; materials and methods; results; limitations, and conclusions. The results of the study should be presented with statistical data.
- The aim of the study is not fully understandable and consistent with the title of the article – as it should be defined straightforwardly, otherwise upon review of the whole article it is still not clear what conclusions might be expected.
- The aim of the study needs-assessment to make it consistent with design of the study.
- - Materials and methods should present the characteristics of the study design (prospective or retrospective study, period of time and others.
- What was the rational of authors to use the terms for literature search? What key words colleagues used?
- Inclusion and exclusion criteria not clear. It should be presented data in exclusion criteria: about age of patients, immunosuppression therapy; diabetes mellitus and autoimmune pathology, TB contact; information about active TB in anamnesis in all persons.
- - The systematic review or meta-analysis review should be carried out in accordance with the PRISMA protocol (http://www.prisma-statement.org), used for this type of study
- Limitations of the study did not present.
- What recommendations are provided colleagues by to medical society?
- Conclusions are not clear (please look also at the first comment regarding the aim of the study).
All these concerns should be well addressed to consider this manuscript suitable for publication.
Author Response
- The title of the article should be corrected.
Response: The article title has been modified.
Publication type must be specified (meta-analisis, review or systematic review
Response: The type of the article is specified as “Article”. This article describes a retrospective review of data in VAERS. The article could also be classified as “Hypothesis”. The type “Article” seems the most appropriate classification of this article, but the authors are open to an alternative article type. The methodologies of meta-analysis review and systematic review do not describe this article.
- The structure of the abstract should be corrected and should include: background; objective; materials and methods; results; limitations, and conclusions.
Response: The abstract was adjusted to align with structured abstract format. The structured abstract headers can be dropped and are left for illustration.
- The results of the study should be presented with statistical data.
Response: Statistical calculations have been added to the manuscript.
- The aim of the study is not fully understandable and consistent with the title of the article – as it should be defined straightforwardly, otherwise upon review of the whole article it is still not clear what conclusions might be expected.
Response: The article title has been modified adding additional details.
- The aim of the study needs-assessment to make it consistent with design of the study.
- - Materials and methods should present the characteristics of the study design (prospective or retrospective study, period of time and others.
Response: The materials and methods section was modified. “This is a retrospective analysis of the VAERS database from January 1, 1990 until October 27, 2023.”
- What was the rational of authors to use the terms for literature search? What key words colleagues used?
Response: Literature relevant to this study was cited as appropriate.
- Inclusion and exclusion criteria not clear. It should be presented data in exclusion criteria: about age of patients, immunosuppression therapy; diabetes mellitus and autoimmune pathology, TB contact; information about active TB in anamnesis in all persons.
Response: Patients with reported adverse events for “Henoch-Schonlein purpura, Kawasaki’s disease, Multisystem inflammatory syndrome, Multisystem inflammatory syndrome in adults, and Multisystem inflammatory syndrome in children, and Vasculitis” were included. No patients were excluded.
- - The systematic review or meta-analysis review should be carried out in accordance with the PRISMA protocol (http://www.prisma-statement.org), used for this type of study
Response: This is a retrospective study, not a systematic review or meta-analysis review.
- Limitations of the study did not present.
Response: Section 4.7 on study limitations has been added.
- What recommendations are provided colleagues by to medical society?
Response: Section 4.8 on study recommendations has been added.
- Conclusions are not clear (please look also at the first comment regarding the aim of the study).
All these concerns should be well addressed to consider this manuscript suitable for publication.
Reviewer 2 Report
Comments and Suggestions for Authors
I cannot understand how you have reached the conclusion that vaccines result in the creation of immune complexes, and this is the cause of those diseases.
Author Response
I cannot understand how you have reached the conclusion that vaccines result in the creation of immune complexes, and this is the cause of those diseases.
Response: Vaccines are intentionally designed to stimulate humoral antibody responses to pathogen protein(s). For primary humoral responses, the authors propose that the resulting antibody titers are too low to trigger KD or MIS; Fc receptors have lower affinity for IgG antibody heavy chain. However, high antibody titer levels are proposed to trigger both KD and MIS. This can be from (1) secondary exposures, (2) persistent infections, (3) addition of maternal antibodies for neonates, etc. By definition immune complexes are antibodies bound to a protein or peptide – the source can be either an infection, immunization, and even environmental exposures to pollens, etc.
Reviewer 3 Report
Comments and Suggestions for Authors
The title is not suitable regarding the paper. The aetyology of Kawasaki disease is too simplified. The ethyology of Kawasaki disease and connection with Covid is well described by Owens AM and. coll. in Stat Pearls (2023)
Author Response
The aetyology of Kawasaki disease is too simplified. The ethyology of Kawasaki disease and connection with Covid is well described by Owens AM and. coll. in Stat Pearls (2023)
Response: Owens & Plewa (2023) provide a nice overview of Kawasaki disease; they indicate that the etiology is “still not known and may be related to a wind-borne or water-borne pathogen”. In this article, we propose that KD is associated with multiple infectious agents as summarized in literature citations and also with immunizations. We propose the key etiology of KD is immune complexes activating Fc receptors on immune cells and platelets. Immune complexes in KD have been previously proposed, see Fossard & Thompson (1977) and review by Menikou et al. (2019).
Round 2
Reviewer 1 Report
Comments and Suggestions for Authors
Dear colleagues,
All my recommendations were added.
Thank you very much. Useful data.
Best wishes,
Anna Starshinova
Reviewer 3 Report
Comments and Suggestions for Authors
I am satisfied wth the improvement of this paper